# STAR: Stability-Inducing Weight Perturbation for Continual Learning

**Masih Eskandar**[1][*], **Tooba Imtiaz**[1], **Davin Hill**[1], **Zifeng Wang**[2][†], **Jennifer Dy**[1]
[1]Department of Electrical & Computer Engineering, Northeastern University
[2] Google Cloud AI Research

## Abstract

Humans can naturally learn new and varying tasks in a sequential manner. Continual learning is a class of learning algorithms that updates its learned model as it sees new data (on potentially new tasks) in a sequence. A key challenge in continual learning is that as the model is updated to learn new tasks, it becomes susceptible to *catastrophic forgetting*, where knowledge of previously learned tasks is lost. A popular approach to mitigate forgetting during continual learning is to maintain a small buffer of previously-seen samples, and to replay them during training. However, this approach is limited by the small buffer size and, while forgetting is reduced, it is still present. In this paper, we propose a novel loss function STAR that exploits the worst-case parameter perturbation that reduces the KL-divergence of model predictions with that of its local parameter neighborhood to promote stability and alleviate forgetting. STAR can be combined with almost any existing rehearsal-based methods as a plug-and-play component. We empirically show that STAR consistently improves performance of existing methods by up to $\sim 15\%$ across varying baselines, and achieves superior or competitive accuracy to that of state-of-the-art methods aimed at improving rehearsal-based continual learning. Our implementation is available at `https://github.com/Gnomy17/STAR_CL`.

## 1 Introduction

Humans are naturally continual learners. They are able to learn different sets of tasks sequentially without significant forgetting or loss of performance at previous tasks. This allows them to learn continuously from many sources of knowledge over time.

In contrast, modern machine learning models have been shown to suffer from *catastrophic forgetting* (McCloskey & Cohen, 1989) during sequential learning of tasks, where the model's performance on previous tasks completely degrades after being trained on a new task. Continual Learning (CL) seeks to solve this problem and enable models to learn consecutively, akin to humans. A large body of work tackles this issue from varying perspectives; these are often inspired by balancing the stability-plasticity dilemma (Mermillod et al., 2013; Mirzadeh et al., 2020a;b), where *plasticity* is the ability to change and learn new knowledge, and *stability* is the ability to preserve previously learned knowledge. Intuitively, these two concepts conflict. The goal of CL is to enforce stability without hindering model plasticity, with limited or no access to the previous training data.

Among different approaches to mitigate catastrophic forgetting, rehearsal-based methods (Aljundi et al., 2018; Chaudhry et al., 2019; Buzzega et al., 2020) have gained a lot of attention due to their simplicity and effectiveness. In rehearsal-based CL, a relatively small buffer of previously seen data is maintained and used in the training of new tasks to alleviate forgetting of previous tasks. Effectively, rehearsal-based CL methods induce stability through replay of buffered samples to the model while training on new data. Many works focus on how to best utilize the limited buffered data effectively to retain the most information about previous data points with access to a limited number of samples.

An under-explored area, however, is the overall stability of the local parameter space when it comes to the model output distribution for previously seen samples. That is, if the

---

[*]Correspondence to eskandar.m@northeastern.edu
[†]Work done while the author was at Northeastern University.

model outputs with respect to previous samples, are stable and unchanging within a local parameter neighborhood, parameter updates in that neighborhood are not going to result in forgetting. Thus, in this work, we attempt to bolster the stability of model predictions in local parameter neighborhoods, so that future parameter updates lead to less forgetting.

We propose a new surrogate measure of forgetting, which is the change in distribution of correct model predictions on buffered samples between now and the future. As future parameters are not accessible in the present, we propose to instead optimize a worst-case version of this measure over a local parameter neighborhood. We achieve this by first perturbing network parameters by a de-stabilizing perturbation, i.e., changing the network outputs as much as possible, and then using the gradient of the perturbed parameters to optimize our objective. Intuitively, this results in leading model parameters into regions where model outputs are more stable locally, and thus gives more flexibility for parameter updates in the future (Figure 1). Our method does not assume any task boundaries or any specific CL method and only requires a memory buffer of previous samples. Thus, we propose our **ST**ability Inducing PA**A**rameter-space **R**egularization (**STAR**) as a plug-and-play component to any existing rehearsal-based CL baseline.

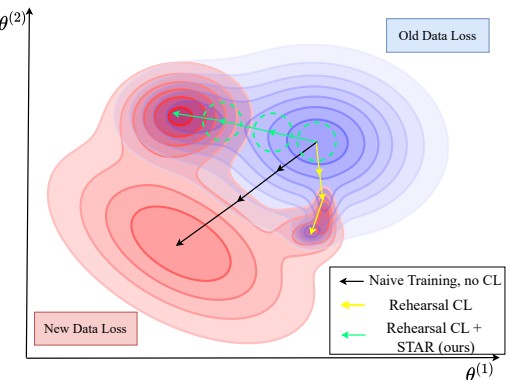

Figure 1: STAR improves rehearsal-based CL by considering the change in output distribution in a local parameter neighborhood. Our regularization objective promotes convergence to regions of the parameter space where the loss is stable over a local neighborhood $\delta$ around the parameters.

Finally, we perform exhaustive experiments, that show that our method leads to significant empirical improvements to various baselines, as well as exploratory studies into its different components. To the best of our knowledge, STAR is the first to exploit worst-case parameter space perturbation for stability in CL. Our contributions are summarized below:

- We present a novel surrogate measure of forgetting, and propose to optimize this measure in a local parameter neighborhood, to increase flexibility of future model updates and reduce forgetting during training.

- STAR assumes no access to task boundaries or task labels during training and only requires a memory buffer. Our method can be combined with existing CL baseline methods with minimal changes.

- We empirically validate the benefits of STAR on multiple existing CL baselines. STAR exhibits significant gains in performance for all evaluated methods across multiple datasets and buffer sizes. Furthermore, we compare the performance improvement of our method with existing baseline-enhancement methods and show superior and occasionally closely competitive performance gain compared to existing methods. Overall we achieve a maximum of **15.11%** increase in average accuracy compared to the baseline across all baselines and datasets.

- We perform exploratory experiments and ablation studies on the different components of our methods and demonstrate their contributions, confirming the importance of our design choices.

## 2 RELATED WORKS

**Continual Learning**   Existing CL methods can generally be categorized as regularization-based, architecture-based, or rehearsal-based approaches. Architecture-based methods (Rusu et al., 2016; Mallya & Lazebnik, 2018; Wang et al., 2020) modify the model architecture by compartmentalizing and expanding the network parameters, keeping the information from new tasks from interfering

catastrophically with the previously learned tasks. Rehearsal-based methods keep a memory buffer of data during training and *replay* these buffered samples to prevent the model from forgetting previous task knowledge. Rehearsal offers a simple and powerful framework to tackle CL and is leveraged in many state-of-the-art methods (Chaudhry et al., 2019; Buzzega et al., 2020). Moreover, an alternative category of CL methods called Regularization-based methods (Kirkpatrick et al., 2017; Zenke et al., 2017; Li & Hoiem, 2017) seeks to ensure model stability through regularization on the weights during optimization. Such methods mainly focus on identifying model weights that are more important to the performance of previous tasks, and impose a penalty on changes to those weights.

In this paper, we focus on improving rehearsal-based methods through regularization using buffered samples. We present STAR, a plug-and-play loss term that can be combined with any general rehearsal-based CL baseline. Among other methods aimed at improving rehearsal-based CL, DualHSIC (Wang et al., 2023) adds a Kernel-based information bottleneck as well as feature alignment to motivate learning of more general and task-invariant features, LiDER (Bonicelli et al., 2022) seeks to induce smoothness through optimization of layer-wise Lipschitz constants of the network, and OCM (Guo et al., 2022) use mutual information maximization to learn task-specific features while keeping the information from previous tasks. In contrast to STAR, OCM and DualHSIC do not consider the change in model outputs in the local parameter neighbourhood and LiDER only considers model Lipschitz constants which can be implicitly related to the landscape of model outputs in a local region but does not consider the input data.

**Weight Perturbation**   Weight perturbation methods have been used in the past to perform variational inference (Khan et al., 2018), speed up training (Wen et al., 2018), and increase generalization of neural networks (Wu et al., 2020; Foret et al., 2021; Lee et al., 2024). The generalization properties and convergence of worst-case weight perturbations has also been studied (Andriushchenko & Flammarion, 2022). In this work, we adopt a worst-case weight perturbation approach in the context of CL as a means to regularize parameter updates across tasks and reduce forgetting. One more related work utilizing weight perturbation is DPCL (Lee et al., 2024), which uses weight perturbation to perform *online* CL. However, in contrast to our work, DPCL uses random noise perturbations by way of stochastic classifiers and does not draw strict relationships to forgetting.

## 3   BACKGROUND AND PRELIMINARIES

Let $f_\theta$ denote a classifier parameterized by $\theta \in \mathbf{R}^P$, where $P$ is the number of network parameters, and $\theta$ represents all network parameters concatenated together. Let $\theta^{(l)}$ denote the parameters of the $l$th layer of the classifier, with a total of $L$ layers. As such, we denote $f_{\theta+\delta}$ as a network whose parameters have been perturbed by some $\delta \in \mathbf{R}^P$. Given an input $x$ belonging to one of $K$ classes, $f_\theta(x)$ denotes the *predicted label* while $q_\theta(x)$ denotes the *output probabilities* of said classifier, i.e. $f_\theta(x) = \arg\max_{k \in \{1,\dots,K\}} q_\theta(x)_k$. Subsequently, as $q_\theta(x)$ represents a categorical probability distribution, we can measure the KL-Divergence between two model output distributions as $\mathcal{KL}(q_\theta(x), q_{\theta'}(x)) = \sum_k q_\theta(x)_k \log \frac{q_\theta(x)_k}{q_{\theta'}(x)_k}$. We denote the parameters at training timestamp $t \in \{0, \dots, \tau\}$ by $\theta_t$; $\theta_{t+s}$ denotes the parameters at timestep $t+s$ for $s \in \{0, \dots, \tau-t\}$. Throughout the paper, we conceptually consider time $t$ to be the present timestamp and $t + s$ to be representative of a timestamp in the future. Finally, the notation $\mathbf{1}_{[condition]}$ denotes the indicator function, which is equal to 1 if the $condition$ holds, and 0 otherwise.

### 3.1   CONTINUAL LEARNING AND REHEARSAL-BASED METHODS

We assume a general supervised CL scenario. Formally, at any given training timestamp $t \in \{0, \dots, \tau\}$, we are given a batch of input-label pairs $(x_t, y_t) \sim \mathcal{X}_t$, where $\mathcal{X}_t$ is the training data distribution at time $t$. We do not make any assumptions on explicit task boundaries, therefore $\mathcal{X}_t$ can change at any time.

The objective is to minimize the classification error on all seen data streams by the end of training. As such, let $\mathcal{X}_{0:t}$ denote a uniform mixture of distributions $\{\mathcal{X}_i\}_{i \le t}$, i.e. any sample drawn from $\mathcal{X}_{0:t}$ is equally likely to be drawn from any $\mathcal{X}_i$. Intuitively, $\mathcal{X}_{0:t}$ is the distribution of all observed data at and before time $t$. Hence, we can write the CL objective as follows:

$$\min_{\theta} \mathbb{E}_{(x,y) \sim \mathcal{X}_{0:\tau}} \left[ \mathbf{1}_{[f_\theta(x) \neq y]} \right] \tag{1}$$

The primary challenge of CL is that the training data distribution $\mathcal{X}_t$ may change over time. Modern learning methods which use stochastic gradient descent often assume no change in data distribution across time; as such, they can encounter catastrophic forgetting when faced with the CL scenario, causing performance to deteriorate significantly on previous data distributions.

More formally, we can quantify forgetting as the difference in error between two timestamps, $t$ and $t + s$ during training (Chaudhry et al. (2019)):

$$Forgetting(t, s) := \mathbb{E}_{(x,y) \sim \mathcal{X}_{0:t}} \left[ \mathbf{1}_{[f_{\theta_{t+s}}(x) \neq y]} - \mathbf{1}_{[f_{\theta_t}(x) \neq y]} \right] \tag{2}$$

**Rehearsal-based Methods.** Rehearsal-based methods (e.g. Chaudhry et al. (2019); Aljundi et al. (2018); Buzzega et al. (2020)) have been shown to help reduce the effects of catastrophic forgetting. These methods maintain a relatively small buffer of previously-seen data samples, and then train jointly on new data samples and buffer samples. We denote the buffer at time $t$ as $\mathcal{M}_t$. It is imperative to note that since the data distribution $\mathcal{X}_t$ changes, so does the buffer, which is why, unlike most previous works, we index it with time $t$.

Thus, the training loss for most rehearsal based CL methods takes the following form

$$\mathcal{L}_{CL}(\theta_t) = \ell(f_{\theta_t}(x_t), y_t) + \ell_{\mathcal{M}_t}(f_{\theta_t}(x_{\mathcal{M}_t}), y_{\mathcal{M}_t}) \tag{3}$$

where $x_t, y_t$ is the current task data, $x_{\mathcal{M}_t}, y_{\mathcal{M}_t}$ are drawn from the buffer, and $\ell, \ell_{\mathcal{M}_t}$ are typically loss functions, such as cross entropy or mean squared error.

**Buffer Management.** The buffer $\mathcal{M}_t$ serves as a surrogate for the distribution $\mathcal{X}_{0:t}$. Since $\mathcal{M}_t$ has limited capacity, sampling strategies are used to balance its distribution. In order to avoid the need for usage of task boundaries during training, mainstream methods such as Aljundi et al. (2018); Buzzega et al. (2020) use the reservoir sampling strategy (Vitter, 1985) to constantly update the buffer while keeping the distribution of data balanced. While we do not make any assumption about the buffer sampling strategy in our method, most state of the art rehearsal methods use the reservoir sampling strategy. We reiterate that although this eliminates the need for task boundaries, it does not result in a static buffer distribution $\mathcal{M}$. Not only do new samples get added and old samples are removed, the distribution of the newly added samples changes with the incoming data distribution.

In this work, we improve the general rehearsal-based loss function $\mathcal{L}_{CL}$ (Equation (3)) by adding a regularization term to improve the stability of previously-learned parameters and further mitigate catastrophic forgetting. Our method can be combined with any rehearsal-based CL method to improve its performance; as such we assume $\mathcal{L}_{CL}$ to be the loss function of some underlying baseline method.

## 4 STABILITY-INDUCING PARAMETER REGULARIZATION

In this section, we propose STAR, a general loss term that can be used to regularize any rehearsal-based CL loss function.

### 4.1 STABILITY LOSS FUNCTION

While $Forgetting$ (eq. (2)) is a useful post-hoc evaluation metric for a trained model, it is impractical to be used during training. In particular, to evaluate $Forgetting(t, s)$ at time $t$ requires access to "future" model parameters at time $t + s$. Instead, in this section we propose a tractable surrogate for estimating forgetting without having access to future parameters $\theta_{t+s}$ or data samples, by way of regularizing the local change in model output distribution with respect to samples and parameters at time $t$. That is, if the model output distribution is unchanging locally (in a neighborhood around $\theta_t$), then parameter updates in any arbitrary direction within this local region will not cause forgetting.

We start with the formulation in eq. (2). For samples on which both of the classifiers are incorrect (i.e. $f_{\theta_t}(x) \neq y \wedge f_{\theta_{t+s}}(x) \neq y$) the two indicator functions cancel out. Hence, we are left with

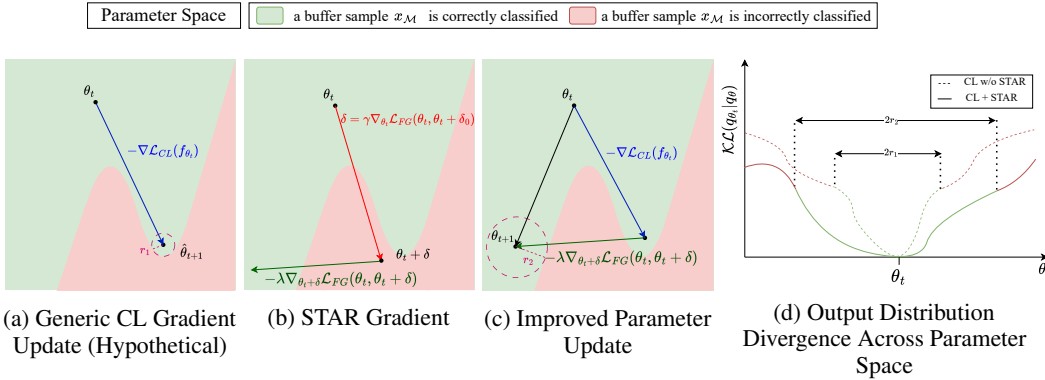

Figure 2: Conceptual illustration of STAR. Vanilla rehearsal-based CL approaches do not consider consistency of output distribution in the local parameter neighborhood (a). By leveraging the gradient of our proposed loss $\mathcal{L}_{STAR}$ (b), we navigate towards parameter regions where the model prediction of previously seen data is consistent (c), reducing forgetting during training due to future parameter updates leading to lower divergence of model output distribution (d).

$$Forgetting(t,s) = \mathbb{E}_{(x,y)\sim\mathcal{X}_{0:t}}\Big[\underbrace{\mathbf{1}_{[f_{\theta_t}(x)=y \,\wedge\, f_{\theta_{t+s}}(x)\neq y]}}_{\text{Forgotten samples}} - \underbrace{\mathbf{1}_{[f_{\theta_t}(x)\neq y \,\wedge\, f_{\theta_{t+s}}(x)=y]}}_{\text{Newly-learned samples}}\Big] \quad (4)$$

The difference in error is comprised of forgotten samples, minus the newly-learned samples. This is why the empirical measure of *forgetting* in eq. (2) can technically be negative. In this work, we focus on reducing the first term, i.e. the error due to the forgotten samples.

Inspired by this decomposition, we propose the change in the distribution of network output probabilities $q_\theta(x)$ over correctly classified samples $x$ s.t. $f_\theta(x) = y$ as a surrogate measure for forgotten samples (first term in eq. (4)). In order to measure the change in distribution, we propose measuring the KL-divergence between output probabilities at time $t$ and $t + s$.

$$\mathbb{E}_{(x,y)\sim\mathcal{X}_{0:t}}\left[\mathbf{1}_{[f_{\theta_t}(x)=y]}\cdot\mathcal{KL}\big(q_{\theta_t}(x)|q_{\theta_{t+s}}(x)\big)\right] \quad (5)$$

In order to optimize eq. (5), we first need access to samples from all previous data distributions. However, we can use the buffer $\mathcal{M}_t$ as a surrogate for $\mathcal{X}_{0:t}$. Therefore, the empirical loss function corresponding to eq. (5) is as follows

$$\mathcal{L}_{FG}(\theta_t,\theta_{t+s}) = \sum_{(x,y)\in\mathcal{M}_t^*}\mathcal{KL}\big(q_{\theta_t}(x)|q_{\theta_{t+s}}(x)\big) \quad (6)$$

$$\mathcal{M}_t^* = \{(x,y)\in\mathcal{M}_t|f_{\theta_t}(x) = y\}$$

The loss function $\mathcal{L}_{FG}$ seeks to promote the prediction of correctly classified samples to stay the same in the future. However, the main issue remains: we do not have access to future parameters $\theta_{t+s}$ at time $t$. We propose to approximate $\theta_{t+s}$ with a local, worst-case perturbation $\theta_t + \delta$, $\delta \in \mathbf{R}^P$, to reduce forgetting regardless of the direction of future parameter updates. We present our novel stability loss function STAR as follows

$$\mathcal{L}_{STAR}(\theta_t) := \max_\delta \mathcal{L}_{FG}(\theta_t,\theta_t+\delta) \quad (7)$$

$$s.t. \|\delta\|_2 \leq d$$

where $d$ is an arbitrary positive number that controls the magnitude of the perturbation neighborhood.

Intuitively, the STAR loss function promotes consistency of model prediction distribution w.r.t. correctly classified samples in a local parameter neighborhood. This gives future model updates more flexibility and robustness *in the parameter space*, making forgetting in the future less likely. A conceptual illustration of this is present in Figure 2: a generic CL gradient update may lead to a point where some current buffer samples are still correctly classified, but are forgotten easily in future parameter updates due to the noise in SGD or samples being replaced in the buffer. In contrast, STAR promotes parameter regions where the output w.r.t. correctly classified samples is consistent.

### 4.2 COMBINED OBJECTIVE FUNCTION

The STAR loss function (eq. (7)) can be combined with any CL method that uses a rehearsal-based loss function in order to reduce forgetting and improve performance. Given some rehearsal-based loss $\mathcal{L}_{CL}$, such as ER (Chaudhry et al., 2019) or its variants, we can use $\mathcal{L}_{STAR}$ as a plug-and-play component and optimize the combined final objective:

$$\mathcal{L}_{final} = \mathcal{L}_{CL}(\theta_t) + \lambda \mathcal{L}_{STAR}(\theta_t) \tag{8}$$

where $\lambda$ is a hyperparameter to control the importance of each objective.

### 4.3 STAR IMPLEMENTATION

We propose to optimize eq. (7) using Stochastic Gradient Descent (SGD). However, optimizing $\mathcal{L}_{STAR}$ is not trivial due to the maximization over $\delta$. In particular, there are two main challenges: 1) calculating the weight perturbation $\delta$, and 2) calculating the gradient with respect to the original parameters $\theta_t$ while the loss depends on $\delta$. Fortunately, these challenges can be addressed by adapting existing works, which leverage worst case weight perturbations in robustness (Wu et al., 2020) and generalization (Foret et al., 2021).

**Optimizing for the Perturbation $\delta$.** Calculating eq. (7) requires computation of $\delta$, which is the locally-maximizing perturbation with respect to the parameters $\theta$. We propose to calculate $\delta$ using gradient ascent, i.e., taking gradient steps to increase the KL-Divergence in eq. (5). Calculating the exact local maximum can be challenging, as $q_\theta$ is typically non-convex. However, as in the case with existing works (Foret et al., 2021; Wu et al., 2020), we find that taking a single maximizing gradient step is sufficient to improve performance in practice. We experiment with multiple gradient steps and present the details in appendix D. Following Li et al. (2018); Wu et al. (2020), instead of using a single constant $r$ as a norm constraint (as shown in eq. (7)), we normalize the gradients of the model layer-by-layer. This is due to the different numeric distributions of the weights of each layer, as well as the scale invariance of the weights; i.e., multiplying by 10 at one layer and dividing by 10 the next layer will lead to the same network.

Therefore, to calculate $\delta$, we take a normalized gradient step (detailed below) towards increasing the KL-Divergence term in eq. (7).

Since $\delta = 0$ is the global minimizer of the KL term, and thus has zero gradient, we initialize $\delta$ as $\delta_0$ with small noise proportional to the layer norm as follows:

$$\delta_0^{(l)} \sim \mathcal{N}(0, \epsilon \|\theta^{(l)}\| I) \tag{9}$$

where $l$ is the $l$th layer index, $\epsilon$ is a small, positive scalar, and $I$ is the identity matrix with appropriate dimensions.

Let $g$ be the gradient of the inner KL term in eq. (7):

$$g := \nabla_{\theta + \delta_0} \mathcal{L}_{FG}(\theta, \theta + \delta_0) \tag{10}$$

Subsequently, we calculate $\delta$ for each layer $l$ in the following manner:

$$\delta^{(l)} := \delta_0^{(l)} + \gamma \frac{\|\theta^{(l)}\|_2}{\|g^{(l)}\|_2} g^{(l)} \tag{11}$$

where $1 \le l \le L$, $\delta^{(l)}$ is the perturbation to layer $l$ parameters $\theta^{(l)}$, and $0 \le \gamma$ is a hyper-parameter to control the perturbation ratio (i.e., $\delta^{(l)}$ is normalized such that $\frac{\|\delta^{(l)}\|_2}{\|\theta^{(l)}\|_2} \approx \gamma$).

**Updating the Parameters $\theta$.** To perform SGD using $\mathcal{L}_{STAR}$ with the $\delta$ calculated in eq. (11), we must now compute the gradient *descent* step of minimizing $\mathcal{L}_{STAR}$ w.r.t. the original parameters $\theta_t$. With the formulation in eq. (11), calculating the gradient of $\mathcal{L}_{FG}$ in eq. (7) with respect to model parameters requires computing the Hessian, or at the very least Hessian-vector products (Foret et al., 2021). However, following Wu et al. (2020); Foret et al. (2021), we approximate the gradient via the gradient measured at the perturbed parameter point, i.e. $\nabla_\theta \mathcal{L}_{FG}(\theta, \theta + \delta) \approx \nabla_{\theta + \delta} \mathcal{L}_{FG}(\theta, \theta + \delta)$, to accelerate computation. It is important to note that this is an approximation, and that $\mathcal{L}_{FG}$ is in

practice non-linear. Otherwise, the minimizing gradient step would be equal to the negative of the maximizing gradient step in eq. (11).

Intuitively, model parameters are first perturbed to a parameter point which results in outputs with high KL-Divergence with outputs of the unperturbed parameters, with respect to correctly classified samples. Subsequently, the value of the STAR loss is equal to the KL-Divergence of the outputs of the perturbed model with the outputs of the unperturbed model. We then compute the gradient of the STAR loss at the perturbed parameter point and use it for SGD. A detailed pseudocode of our training algorithm can be found in Algorithm 1.

---

**Algorithm 1** STAR Enhanced Rehearsal-Based CL Pseudocode

---

**Require:** Data stream $\mathcal{X}_t$, parameters $\theta$ with $L$ layers, scalar $\lambda$, perturbation coefficient $\gamma$, learning rate $\alpha$, memory buffer $\mathcal{M}$, CL loss function $\mathcal{L}_{CL}$

$\quad \mathcal{M}_0 \leftarrow \{\}$
$\quad$ Initialize $\theta_0$
$\quad$ **for** $0 \leq t < \tau$ **do**
$\quad\quad x_t, y_t \sim \mathcal{X}_t$
$\quad\quad x_{\mathcal{M}_t}, y_{\mathcal{M}_t} \leftarrow \mathcal{M}_t$
$\quad\quad x^* \leftarrow \{x_{\mathcal{M}_t} | f_{\theta_t}(x_{\mathcal{M}_t}) = y_{\mathcal{M}_t}\}$ $\qquad\qquad\qquad$ ▷ Correctly classified buffer samples
$\quad\quad$ **for** $1 \leq l \leq L$ **do** $\qquad\qquad\qquad\qquad$ ▷ Calculate perturbation for each layer
$\quad\quad\quad \delta_0^{(l)} \sim \mathcal{N}(0, \epsilon \|\theta_t^{(l)}\|_2 I)$
$\quad\quad\quad g^{(l)} \leftarrow \nabla_{\theta_t + \delta_0} \mathcal{KL}(q_{\theta_t}(x^*), q_{\theta_t + \delta_0}(x^*))$ $\qquad\qquad\qquad$ ▷ $\mathcal{L}_{FG}$ Gradient
$\quad\quad\quad \delta^{(l)} \leftarrow \delta_0^{(l)} + \gamma \frac{\|\theta_t^{(l)}\|_2}{\|g^{(l)}\|_2} g^{(l)}$
$\quad\quad$ **end for**
$\quad\quad u_t \leftarrow \lambda \nabla_{\theta_t + \delta} \mathcal{KL}(q_{\theta_t}(x^*), q_{\theta_t + \delta}(x^*))$ $\qquad\qquad\qquad$ ▷ STAR gradient
$\quad\quad u_t \leftarrow u_t + \nabla_{\theta_t} \mathcal{L}_{CL}(x_t, y_t, f_{\theta_t}, \mathcal{M}_t)$ $\qquad\qquad$ ▷ Generic CL method gradient
$\quad\quad \theta_{t+1} \leftarrow \theta_t - \alpha u_t$ $\qquad\qquad\qquad$ ▷ Update parameters with both gradients
$\quad\quad \mathcal{M}_{t+1} \leftarrow sample(\mathcal{M}_t, x_t, y_t)$
$\quad$ **end for**
$\quad$ **return** $\theta_\tau$

---

## 5 EXPERIMENTS

To evaluate the effectiveness of our method, we conduct comprehensive experiments on various CL benchmarks. We adhere to the challenging class-incremental CL scenario, where the task identity of samples is not known during inference, and the model uses a single classification head for all tasks. We incorporate STAR as a plug-and-play component into various SOTA CL frameworks to demonstrate significant improvement, and also compare against existing plug-and-play SOTA CL methods. Additionally, we perform exploratory experiments and an ablation study on the different components of our method.

### 5.1 EXPERIMENTAL SETTING

**Datasets.** We evaluate STAR on three mainstream CL benchmark datasets. **Split-CIFAR10** and **Split-CIFAR100** are the CIFAR10/100 datasets (Krizhevsky et al., 2009), split into 5 disjoint tasks of 2 and 20 classes respectively. **Split-miniImagenet** is a subsampled version of the Imagenet dataset (Deng et al., 2009), split into 20 disjoint tasks of 5 classes each. Both CIFAR10 and CIFAR100 are comprised of 32x32 input images, while MiniImagenet uses 84x84 input sizes. Following previous works (Bonicelli et al., 2022; Wang et al., 2023), for CIFAR10-100 we use a batch size of 32 and 50 epochs per task. For miniImagenet we use a batch size of 128 and 80 epochs per task. For a full list of hyperparameters, see appendix F.

**Comparing Baselines.** STAR is a plug-and-play CL framework that can be combined with any rehearsal-based CL method. We combine STAR with ER(Chaudhry et al., 2019), ER-ACE Caccia et al. (2022), X-DER-RPC (Boschini et al., 2022) and DER++ (Buzzega et al., 2020) as SOTA rehearsal-based methods, to demonstrate the gains in performance.

Additionally, following Bonicelli et al. (2022), we compare with existing SOTA regularization-based methods applied as enhancements to rehearsal-based methods. We use ER-ACE and DER++ as representative baselines and add each method to both and compare the performance improvements. We compare to sSGD (Mirzadeh et al., 2020b), oEwC (Schwarz et al., 2018), OCM(Guo et al., 2022), LiDER (Bonicelli et al., 2022) and DualHSIC (Wang et al., 2023). Naturally, we would prefer to also compare against DPCL (Lee et al., 2024). However, the experimental setting in their paper is the online CL setting (which is different than ours). Furthermore, they do not open-source their implementation, thus we cannot compare against DPCL in a fair and reasonable fashion.

Finally, for reference, we include the **Sequential** baseline which consists of performing CL training without any enhancements to facilitate CL, such as rehearsal or regularization. We also include a potential upper bound on performance, termed **Joint**, which consists of training with access to the all of the training data in an i.i.d. manner.

**Evaluation Metrics.** Following previous works Chaudhry et al. (2019); Lopez-Paz & Ranzato (2017); Wang et al. (2023), we report average accuracy in all tables, and refer readers to appendix B for final forgetting values. The formal definitions of these metrics, as well as information about error bars and number of runs are presented in appendix A.

## 5.2 INTEGRATING STAR WITH REHEARSAL-BASED CL BASELINES

In this section, we empirically evaluate the effect of integrating STAR with various rehearsal-based CL baselines. STAR can be combined with most SOTA baselines to improve their performance. We present the results in Table 1 and observe that STAR consistently improves performance across different methods and rehearsal buffer sizes. Notably, even simple baselines like ER benefit from integrating STAR, as well as more complex methods like X-DER, which incorporates numerous enhancements to the baseline rehearsal method. Therefore, including the STAR regularization improves performance for a wide range of rehearsal-based methods.

Table 1: Comparison of adding STAR to baseline rehearsal-based CL methods in terms of average accuracy.

| Method | Split-Cifar10 | | | Split-Cifar100 | | | Split-miniImageNet | | |
|---|---|---|---|---|---|---|---|---|---|
| Sequential | 19.67 | | | 9.29 | | | 4.51 | | |
| Joint | 92.38 | | | 73.29 | | | 53.55 | | |
| Buffer Size | 100 | 200 | 500 | 200 | 500 | 2000 | 1000 | 2000 | 5000 |
| ER | 36.39 | 44.79 | 57.74 | 14.35 | 19.66 | 36.76 | 8.37 | 16.49 | 24.17 |
| + STAR (ours) | **51.5** | **59.3** | **70.70** | **19.64** | **29.64** | **44.65** | **11.83** | **16.64** | **25.83** |
| ER-ACE | 52.95 | 61.25 | 71.16 | 29.22 | 38.01 | 49.95 | 17.95 | 22.60 | 27.92 |
| + STAR (ours) | **60.69** | **67.58** | **75.44** | **30.38** | **40.20** | **51.67** | **21.06** | **24.9** | **31.01** |
| DER++ | 57.65 | 64.88 | 72.70 | 25.11 | 37.13 | 52.08 | 18.02 | 23.44 | 30.43 |
| + STAR (ours) | **61.76** | **68.60** | **76.52** | **27.64** | **39.77** | **53.24** | **22.4** | **28.19** | **33.36** |
| X-DER (RPC) | 52.75 | 58.48 | 64.77 | 37.23 | **48.53** | 57.00 | 23.19 | 26.38 | 29.91 |
| + STAR (ours) | **58.85** | **65.94** | **69.19** | **38.15** | 47.56 | **57.55** | **24.6** | **27.95** | **32.6** |

## 5.3 COMPARISON WITH OTHER ENHANCEMENTS OF REHEARSAL-BASED METHODS

Having established that STAR can be combined with various baselines to improve performance, we now evaluate STAR against existing approaches that seek to enhance rehearsal-based baselines. The results are presented in Table 2. STAR achieves superior performance in most scenarios and is closely competitive in all other settings. In particular, STAR performs exceptionally well with smaller buffer sizes. This is in line with our intuition: With smaller buffer sizes, fewer samples are retained in the buffer and replayed in future time steps. However, STAR promotes stability in the output distribution across future updates regardless of whether or not a sample is still in the buffer in the future. Hence, it is able to achieve considerable performance boosts.

Table 2: Comparison with other regularization methods in average accuracy, our results are highlighted. The best results are bold, and second-best are underlined.

| Method | | Split-Cifar10 | | | Split-Cifar100 | | | Split-miniImageNet | | |
|---|---|---|---|---|---|---|---|---|---|---|
| Buffer Size | | 100 | 200 | 500 | 200 | 500 | 2000 | 1000 | 2000 | 5000 |
| ER-ACE | | 53.90 | 63.41 | 70.53 | 26.28 | 36.48 | 48.41 | 17.95 | 22.60 | 27.92 |
| +sSGD | | 56.26 | 64.73 | 71.45 | 28.07 | 39.59 | 49.70 | 18.11 | 22.43 | 24.12 |
| + oEWC | | 52.36 | 61.09 | 68.70 | 24.93 | 35.06 | 45.59 | 19.04 | 24.32 | 29.46 |
| +OCM | | 57.18 | 64.65 | 70.86 | 28.18 | 37.74 | 49.03 | 20.32 | 24.32 | 28.57 |
| +LiDER | | 56.08 | 65.32 | 71.75 | 27.94 | 38.43 | 50.32 | 19.69 | 24.13 | 30.00 |
| +DualHSIC | | 57.03 | 64.05 | 72.35 | **30.97** | 39.73 | 49.94 | 20.75 | **25.65** | **31.16** |
| + STAR (ours) | | **60.69** | **67.58** | **75.44** | 30.38 | **40.20** | **51.67** | **21.06** | 24.9 | 31.01 |
| DER++ | | 57.65 | 64.88 | 72.70 | 25.11 | 37.13 | 52.08 | 18.02 | 23.44 | 30.43 |
| + sSGD | | 55.81 | 64.44 | 72.05 | 24.76 | 38.48 | 50.74 | 16.31 | 19.29 | 24.24 |
| + oEWC | | 55.78 | 63.02 | 71.64 | 24.51 | 35.22 | 51.53 | 18.87 | 24.53 | 31.91 |
| +OCM | | 59.25 | 65.81 | 73.53 | 27.46 | 38.94 | 52.25 | 20.93 | 24.75 | 31.16 |
| +LiDER | | 58.43 | 66.02 | 73.39 | 27.32 | 39.25 | 53.27 | 21.58 | **28.33** | **35.04** |
| +DualHSIC | | 58.90 | 67.11 | 74.34 | 22.44 | 39.16 | **54.07** | 21.73 | 27.42 | 33.74 |
| + STAR (ours) | | **61.76** | **68.60** | **76.52** | **27.64** | **39.77** | 53.24 | **22.4** | 28.19 | 33.36 |

## 5.4 Ablation and Exploratory Experiments

**Ablation Study.** We explore the contributions of different components of STAR towards the overall performance. Specifically, we study the effect of two key components of our loss function eq. (7):

- $x^*$ vs. $x$: Using only correctly classified buffer samples $x^*$ versus using all buffer samples $x$ in the STAR loss function.
- $\nabla$ vs. $z$: Perturbing the weights by the gradient of our forgetting surrogate in eq. (5) $\nabla$ versus a random perturbation of the same magnitude by replacing $\delta$ in eq. (11) with Gaussian noise $z$, i.e., $\delta'^{(l)} = \gamma \frac{\|\theta^{(l)}\|_2}{\|z^{(l)}\|_2} z^{(l)}, \quad z^{(l)} \sim \mathcal{N}(0, I)$.

We study the contribution of each component towards the final performance by conducting a series of experiments adding one component at a time while keeping other hyper-parameters fixed. The results on the S-CIFAR10 dataset are presented in Table 3 and show that each component is important to the final performance increase across different buffer sizes.

Notably, the biggest gain in performance corresponds to using the gradient of the stability loss for weight perturbation, as opposed to a random perturbation of the same magnitude (Table 3 rows two and three vs the last two rows). This aligns with our intuition, as using the gradient corresponds to optimizing the worst-case scenario which acts as an upper bound on the local neighborhood. This increases the flexibility for future parameter updates. On the other hand, a random perturbation only improves the average in the local neighborhood.

**Visualizing our Forgetting Surrogate.** In this section we perform an experiment to validate our claim that our proposed STAR loss function eq. (7) is indeed optimizing our forgetting surrogate eq. (5) in the local neighborhood. In section 4 we noted that eq. (5) relies on parameters from the future and thus cannot be optimized directly during training. However, we can conduct a post-hoc exploratory experiment and directly measure $\mathcal{L}_{FG}$ (eq. (5)) empirically after training is complete as we have access to parameters across training. For this purpose, let $\theta_{t_i}$ be the model parameters immediately after training on the $i$th task in the S-CIFAR10 dataset. For each task $i$, we measure $\mathcal{L}_{FG}(\theta_{t_i}, \theta_t)$ for $t > t_i$ on the test set for task $i$. Moreover, to verify we are optimizing the worst-case of our surrogate as well, we measure $\mathcal{L}_{FG}(\theta_{t_i}, \theta'_t)$ as well, where $\theta'_t$ is $\theta_t$ after 5 gradient ascent steps as detailed in section 4.3. We repeat the experiment with and without including STAR and plot the results in Figure 3. The results show a significant decrease in divergence of future model output distributions when using STAR, as expected.

**Choice of Buffer Data or Current Task Data.** In eq. (6), we use data from the buffer as a surrogate for all past distributions. However, one can argue that adding incoming current task data to the mix

can improve the consistency of model predictions for the current task and further reduce forgetting. In Table 4, we experiment with using current task data in the STAR loss function. We find that combining both buffer and current task data does not lead to any significant improvements, and using only current task data causes a significant increase in forgetting. We hypothesize two reasons as to why. Firstly, if we only use current task data and not buffer data, once the data distribution shift occurs, the consistency of predictions for previous data is no longer considered. This shift in objective leads to a shift in the corresponding minima for the rehearsal loss, reducing stability significantly. Secondly, our objective seeks to keep the overall output distribution of already learned samples consistent. However, in contrast to buffered samples, the output distribution of current samples are not fully trained and therefore keeping them consistent does not necessarily benefit performance.

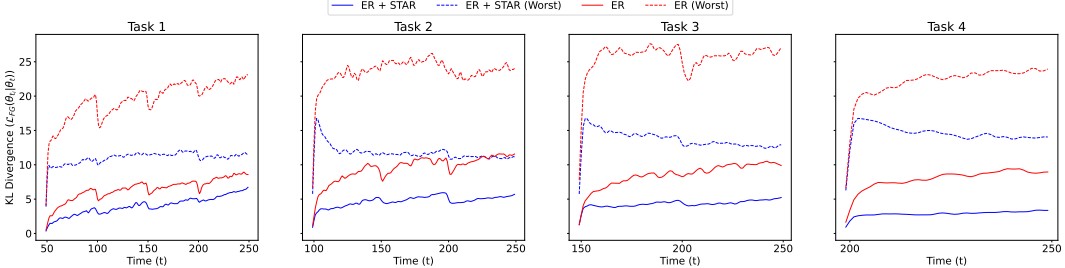

Figure 3: KL Divergence of the correctly classified samples of the test set of each task for the S-CIFAR10 Dataset, between model predictions immediately after training on that task, and future parameter points during training. "Worst" indicates the KL Divergence after multiple gradient ascent steps as detailed in section 4.3.

Table 3: Effect of method components on average accuracy. The second column represents whether only correctly classified samples ($x^*$) or all samples ($x$) were used in STAR, the third column represents whether the weight perturbation for calculating the loss is $\nabla \mathcal{L}_{FG}$ or random noise ($z$). All other hyper-parameters are fixed.

| Method | $x^*$ vs. $x$ | $\nabla_\theta$ vs. $z$ | 100 | 200 | 500 |
|---|---|---|---|---|---|
| DER++ | - | - | 55.36 | 62.97 | 70.48 |
| + STAR | $x$ | $z$ | 56.28 | 63.15 | 72.38 |
| | $x^*$ | $z$ | 56.20 | 66.82 | 72.51 |
| | $x$ | $\nabla_\theta$ | 60.83 | 67.03 | 75.11 |
| | $x^*$ | $\nabla_\theta$ | **61.76** | **68.06** | **76.52** |

Table 4: Effect of using current task data for $\mathcal{L}_{STAR}$ as measured in average accuracy. The highlighted row represents the proposed setting for STAR.

| Method | Buffer | Current | 100 | 200 | 500 |
|---|---|---|---|---|---|
| DER++ | - | - | 57.52 | 65.07 | 73.05 |
| + STAR | ✗ | ✓ | 24.25 | 30.29 | 34.06 |
| | ✓ | ✗ | **61.76** | **68.60** | **76.52** |
| | ✓ | ✓ | 58.88 | 52.55 | 75.48 |

## 6 CONCLUSION

The principal challenge of Continual Learning is overcoming catastrophic forgetting. Inspired by stability-plasticity trade-off, we propose using the divergence in model output distribution for correctly classified samples as a novel surrogate for forgetting. Subsequently, we introduce STAR, which is a regularizer for boosting the performance of rehearsal-based continual learners. STAR exploits the worst-case parameter perturbation that reduces the KL-divergence of model predictions with that of its local parameter neighborhood to promote stability and alleviate forgetting. Our loss function can be used as a plug-and-play component to improve any rehearsal-based CL method. We demonstrate the superior performance of STAR through extensive experiments on various CL benchmarks against state-of-the-art continual learners. Our novel proposed formulation and methodology provides a new perspective for CL research via optimizing the change in model prediction distribution across the local parameter-space.

## ETHICS STATEMENT

STAR is a plug-and-play enhancement that can be integrated into almost any rehearsal-based CL method. Therefore, we should address the baseline CL method used for STAR in terms of fairness and bias (Mehrabi et al., 2021) as it could extend into the resulting model from STAR. Moreover, extra measures should be taken when it comes to user data privacy (Al-Rubaie & Chang, 2019) or robustness to adversaries (Madry et al., 2018) in safety-critical applications.

## REPRODUCIBILITY STATEMENT

We present thorough explanations of implementation details for STAR in section 4.3, as well as a detailed pseudo-code in algorithm 1. Moreover, we present details of our experimental settings in section 5.1 and appendix A, as well all the used hyper-parameters in table 8. Furthermore, we have made our implementation publicly available on Github[1].

## ACKNOWLEDGEMENTS

This work was partially supported by NIH/NHLBI 1RO1HL171213-01A1, NIH/NCI U24CA264369, NIH/NCI R01CA240771, NIH 2T32HL007427- 41, and DoD/USAMRAA HT94252410553. We want to thank all ICLR reviewers and area-chairs for their efforts toward improving this work.

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

## SUPPLEMENTARY MATERIAL

## A  ADDITIONAL EXPERIMENTAL DETAILS

**Definition of Evaluation Metrics**  Assuming the dataset has $k$ disjoint tasks, average accuracy $A_\tau$ and final forgetting $F_\tau$ at final training timestep $\tau$ can be computed as follows

$$A_\tau = \frac{1}{k} \sum_{i=1}^{k} a_{i,\tau} \tag{12}$$

$$F_\tau = \frac{1}{k-1} \sum_{i=1}^{k-1} \max_{j \in \{1,\dots,k-1\}} [a_{i,t_j} - a_{i,\tau}] \tag{13}$$

where $a_{i,t}$ is the accuracy on the $i$th task test set evaluated on model parameterized by $\theta_t$ and $t_j$ is the timestamp after training on data from the $j$th task.

For each experiment, we averaged over 5 runs with different seeds. The error bars are presented in Table 6. All experiments were run on a single Nvidia RTX A6000 GPU.

Following previous works (Bonicelli et al., 2022), we use the ResNet18 (He et al., 2016) architecture for CIFAR10 and CIFAR100 datasets and the efficient-net b2 (Tan, 2019) architecture for miniImagenet.

## B  FORGETTING VALUES

We present forgetting values in Table 5. Consistent with the increase in average accuracy when using STAR, we observe that the forgetting is comparable or significantly reduced when STAR is integrated with the baseline CL objectives.

| Method | Split-Cifar10 | | | Split-Cifar100 | | | Split-miniImageNet | | |
|---|---|---|---|---|---|---|---|---|---|
| Buffer Size | 100 | 200 | 500 | 200 | 500 | 2000 | 1000 | 2000 | 5000 |
| ER | 55.90 | 44.46 | 38.15 | **70.17** | 63.92 | 46.56 | **63.55** | **54.14** | **41.32** |
| + STAR (ours) | **42.03** | **36.22** | **20.39** | 72.54 | **61.76** | **41.97** | 69.41 | 62.97 | 49.76 |
| ER-ACE | 22.76 | **18.30** | 14.96 | 50.63 | 38.21 | 27.90 | 29.82 | **23.74** | **19.72** |
| + STAR (ours) | **21.01** | 19.32 | **12.03** | **39.50** | **34.45** | **24.62** | **26.70** | 25.11 | 20.38 |
| DER++ | 40.25 | 30.06 | 21.85 | 62.92 | 49.80 | 31.10 | 63.40 | 46.69 | 37.11 |
| + STAR (ours) | **33.87** | **21.29** | **17.91** | **52.94** | **40.46** | **27.64** | **36.59** | **26.77** | 27.47 |
| X-DER (RPC) | 18.72 | 15.16 | 12.29 | 41.58 | 31.84 | 17.01 | 48.67 | 38.33 | 28.29 |
| + STAR (ours) | **17.38** | **13.36** | **10.46** | **29.79** | **26.10** | **14.46** | **27.36** | **22.37** | **16.78** |

Table 5: Final Forgetting (FF) values for Table 1. Lower is better.

| Method | Split-Cifar10 | | | Split-Cifar100 | | | Split-miniImageNet | | |
|---|---|---|---|---|---|---|---|---|---|
| Buffer Size | 100 | 200 | 500 | 200 | 500 | 2000 | 1000 | 2000 | 5000 |
| ER * | 1.23 | 1.64 | 0.97 | 0.64 | 1.56 | 0.94 | 1.11 | 0.93 | 1.46 |
| + STAR (ours) | 6.28 | 4.90 | 2.16 | 1.01 | 1.13 | 3.52 | 0.28 | 0.56 | 0.16 |
| ER-ACE* | 0.88 | 0.79 | 0.99 | 1.12 | - | - | 0.43 | - | - |
| + STAR (ours) | 1.64 | 1.41 | 0.54 | 1.69 | 0.57 | 0.14 | 0.42 | 0.53 | 0.14 |
| DER++* | 1.90 | 0.91 | 0.53 | 1.32 | - | - | 1.42 | - | - |
| + STAR (ours) | 1.95 | 1.10 | 0.94 | 0.51 | 0.59 | 1.22 | 1.77 | 1.07 | 0.55 |
| X-DER (RPC)* | 0.76 | 1.49 | 2.07 | 2.26 | - | - | 0.73 | - | - |
| + STAR (ours) | 1.66 | 1.15 | 1.22 | 0.61 | 0.46 | 0.27 | 0.40 | 0.93 | 0.57 |

Table 6: Standard deviation of the average accuracies reported in Table 1. '*' means the result was taken from existing works, '-' means the error bar was not provided in the original work

## C RUNNING TIME

STAR uses worst-case weight perturbation to improve rehearsal-based continual learning. While STAR requires additional computation, namely two extra optimization steps per batch, we believe this cost is not prohibitive and it is natural to have a trade-off of performance and computation cost. For the sake of completeness, we report the running time of the experiments of Table 1 in Table 7 below.

| Method | Split-Cifar10 | Split-Cifar100 | Split-miniImageNet |
|---|---|---|---|
| ER * | 47 | 52 | 142 |
| + STAR (ours) | 106 | 77 | 305 |
| ER-ACE* | 49 | 57 | 146 |
| + STAR (ours) | 87 | 112 | 296 |
| DER++* | 71 | 76 | 259 |
| + STAR (ours) | 124 | 109 | 372 |
| X-DER (RPC)* | 64 | 77 | 241 |
| + STAR (ours) | 107 | 130 | 396 |

Table 7: Running time of methods (mins) for Table 1.

## D NUMBER OF STEPS

We experiment with taking multiple gradient steps as opposed to one, for calculating the weight perturbation in eq. (7). We present the results in fig. 4, and show that taking three gradient steps can

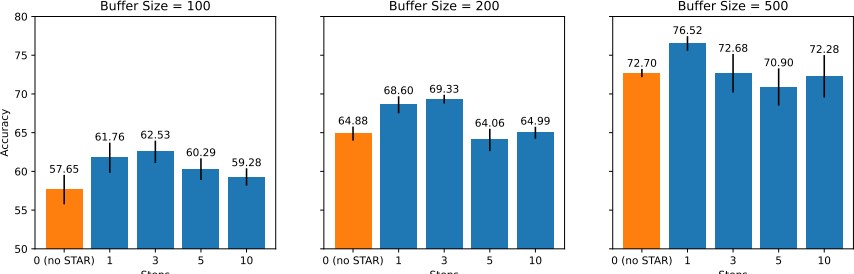

Figure 4: Effect of Number of Gradient Steps in the Maximization Objective on Performance as Measured in Average Accuracy

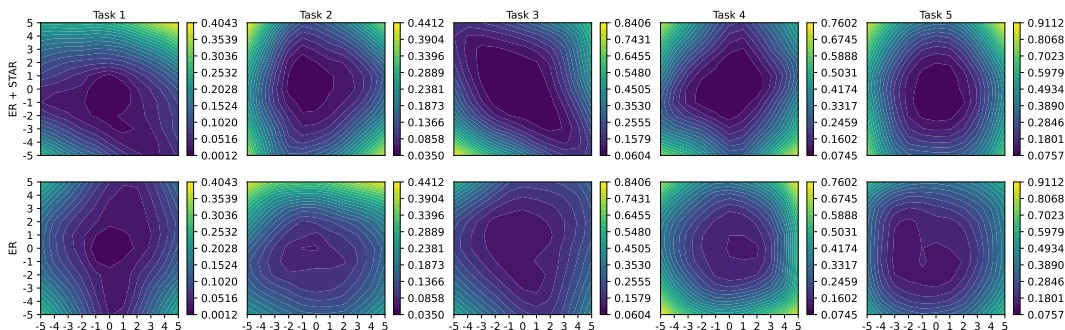

Figure 5: Contour lines of the cross entropy loss values, on the set of tasks seen so far (darker is lower). STAR leads to larger low loss regions.

sometimes lead to a slight increase in performance, at the cost of extra computation. Additionally, we see that beyond that leads to a reduction in the performance gain. We hypothesize that this is due to the gradient approximation discussed in section 4.3. As we take more and more optimization steps, the gradient approximation becomes less accurate, leading to less improvements.

## E    EMPIRICAL VISUALIZATION OF THE LOSS LANDSCAPE

Figure 1 and Figure 2 are conceptual illustrations meant to convey the intuition of our methodology. For the sake of completeness, we conduct an empirical visualization of the cross-entropy loss landscape. While training on the S-CIFAR10 dataset, we record the weights of the network after training on each task and visualize the loss on the test set of the tasks seen so far. To visualize the loss, we perturb the network along two random (multivariate standard Gaussian) directions, normalized in the same scheme as detailed in section 4.3. We repeat the same experiment on ER with and without applying STAR while starting from the same seed, and present the visualization in Figure 5. We can see that applying STAR leads to loss minima with a larger low-loss region, leading to more "flexibility" for future parameter updates.

## F    HYPER-PARAMETERS FOR STAR

We find our hyperparameters using grid-search on a validation set. We present our hyperparameters in table 8. For any missing hyperparameters, we refer the readers to the works of Buzzega et al. (2020); Bonicelli et al. (2022); Boschini et al. (2022) as we used the same values.

| Dataset | Seq-CIFAR10 | | | Seq-CIFAR100 | | | Seq-miniImageNet | | |
|---|---|---|---|---|---|---|---|---|---|
| Buffer Size | 100 | 200 | 500 | 200 | 500 | 2000 | 1000 | 2000 | 5000 |
| ER + STAR | | | | | | | | | |
| lr | 0.1 | 0.1 | 0.1 | 0.1 | 0.1 | 0.05 | 0.1 | 0.1 | 0.1 |
| $\gamma_{STAR}$ | 0.01 | 0.01 | 0.01 | 0.05 | 0.05 | 0.05 | 0.01 | 0.05 | 0.01 |
| $\lambda_{STAR}$ | 0.1 | 0.1 | 0.1 | 0.05 | 0.05 | 0.05 | 0.1 | 0.1 | 0.1 |
| ER-ACE + STAR | | | | | | | | | |
| lr | 0.03 | 0.03 | 0.03 | 0.05 | 0.05 | 0.05 | 0.1 | 0.1 | 0.1 |
| $\gamma_{STAR}$ | 0.01 | 0.01 | 0.01 | 0.05 | 0.05 | 0.05 | 0.01 | 0.05 | 0.01 |
| $\lambda_{STAR}$ | 0.1 | 0.05 | 0.1 | 0.05 | 0.05 | 0.05 | 0.1 | 0.1 | 0.1 |
| DER++ + STAR | | | | | | | | | |
| lr | 0.1 | 0.1 | 0.1 | 0.03 | 0.03 | 0.03 | 0.1 | 0.1 | 0.1 |
| $\gamma_{STAR}$ | 0.05 | 0.05 | 0.05 | 0.03 | 0.03 | 0.03 | 0.01 | 0.01 | 0.005 |
| $\lambda_{STAR}$ | 0.05 | 0.05 | 0.05 | 0.1 | 0.1 | 0.1 | 0.1 | 0.1 | 0.1 |
| $\alpha$ | 0.15 | 0.15 | 0.15 | 0.2 | 0.4 | 0.4 | 0.3 | 0.2 | 0.3 |
| $\beta$ | 0.15 | 0.15 | 0.15 | 0.3 | 0.2 | 0.3 | 0.1 | 0.1 | 0.4 |
| XDER-RPC + STAR | | | | | | | | | |
| lr | 0.03 | 0.03 | 0.03 | 0.03 | 0.03 | 0.03 | 0.1 | 0.1 | 0.1 |
| $\gamma_{STAR}$ | 0.001 | 0.001 | 0.001 | 0.05 | 0.01 | 0.01 | 0.001 | 0.001 | 0.001 |
| $\lambda_{STAR}$ | 0.01 | 0.01 | 0.01 | 0.05 | 0.05 | 0.05 | 0.01 | 0.01 | 0.01 |

Table 8: Choice of hyperparameters for combination of STAR and CL baselines.

