# OpenReview forum: "STAR: Stability-Inducing Weight Perturbation for Continual Learning"
_ICLR.cc/2025/Conference — ICLR 2025 Poster_

### Official Review · Reviewer_Rtfe · 2024-10-29

**Soundness:** 2
**Presentation:** 3
**Contribution:** 2
**Rating:** 5
**Confidence:** 4

**Summary:**

The paper proposes a new method ‘STAR’ to prevent catastrophic forgetting in continual learning, by minimizing the KL divergence between the output of the current parameters and the worst case parameters in a neighborhood of the current ones. The hypothesis is that if the parameters in the neighborhood of the current solution don’t change the output much for past tasks, then it will be easier to find a solution within that region that also performs well on new tasks. Since the exact computation of this method is intractable, a practical approximation is proposed by first taking a gradient ascent step to find the worst case parameters and then it is assumed that the gradient at the worst case parameters is approximately equal to the gradient of the actual current parameters. The approach is tested by combining this idea with several state of the art rehearsal methods. STAR consistently improves other rehearsal baselines across different datasets and memory sizes.

**Strengths:**

* The paper is well written and clearly explains the followed methodology
* The hypothesis and solution are plausible
* The results are well tested with regard to improvement of typical CL baselines and adequately compared to other similar approaches.
* Section 3 is especially clear, which makes the remainder of the paper a lot easier to understand.

**Weaknesses:**

* The main hypothesis of this paper is that it is important to reduce the difference in output with the worst case parameters in the neighborhood of the current solution. A loss function is proposed to avoid the worst case parameters, but I am not convinced that is sufficiently shown that this works as intended. Figure 3 does show that the final KL divergence between the current model and the final model are reduced, but that doesn’t imply anything about the worst case situation, only that one specific instance in the neighborhood is closer. To test this, an experiment could be done were a gradient ascent step is taken as in Equation 10 for both the proposed solution and one of the other replay benchmarks to directly compare the worst case parameters. An alternative explanation for the current results may be that the additional loss function acts as a good regularizer to prevent overfitting on the memory samples.

* The drawing in Figure 1 and 2 are solely hypothetical. There is no evidence that the actual loss landscape looks like this nor that the proposed method actually follows the path that is indicated in these figures. Without evidence that this is the problem, it is hard to accept a solution as long as the problem is not clearly identified.

* The mathematical derivation in lines 288:321 is confusing. First a gradient ascent step is taken to maximize the KL divergence (Eq. 10). Then at those parameters a new gradient is calculated to minimize the same KL divergence (line 323), which should be equal to the negative gradient of Eq. 10, if linearity is assumed (which is done in line 323). Applying this gradient at parameters $\theta$ is then simply a gradient descent step at the initial parameters. So either this derivation could be simplified, or it could be shown that because of the non-linearity of the loss surface, these extra steps actually make a difference (but then the assumption in line 323 is no longer accurate). Figure 2 shows this differently, but only because non-linearity is assumed there.

* Line 051: in a class incremental setting stability is sometimes not sufficient; if new classes are similar to representation of old classes may need to change too. (E.g. if a model had only learned a color representation of a past object and later a new yellow object is added an old yellow object cannot be represented as only being yellow).

 * Line 109: repeated sentence from earlier.

**Questions:**

* Is it possible that the results are explained by a different hypothesis, e.g. reduced overfitting on the memory?
* Is there any empirical evidence for the loss landscapes in Figures 1 and 2?
* Can the mathematical derivation in 4.3 be simplified, or the importance of non-linearities be highlighted?
* Is local stability always sufficient in a class incremental settings, as is claimed in line 051?

---

> ### Author Response · Authors · 2024-11-15
> **Initial Response**
>
> Thank you for your constructive feedback and for appreciating our proposed solution, evaluation, and the clarity of presentation. Regarding your feedback:
>
> **Figure 3**: We thank the reviewer for the suggestion on measuring the KL Divergence after a gradient ascent step. We think that this does indeed further verify our claims, and we have added the modified experiment to the revised pdf (Fig. 3). We take 5 maximizing gradient ascent steps, as detailed in sec 4.3, and plot the KL Divergence. For comparison, we also plot the KL Divergence without applying any maximizing steps. We observe that STAR reduces the KL divergence in both scenarios.
>
> **Smoothness of loss landscape**: Figures 1 & 2 are indeed hypothetical; we use them to illustrate the intuition behind STAR and to enhance the clarity of the method description, as the reviewer kindly noted. We thank the reviewer for the suggestion of  empirically measuring the smoothness of the local loss landscape. We will include it in the camera-ready version.
>
> **Mathematical derivation**: We thank the reviewer for pointing out the confusion regarding the linearity approximation. “It is important to note that this is an approximation, and that $\mathcal{L}_{FG}$ is, in practice,non-linear. Otherwise, the minimizing gradient step would be equal to the negative of the maximizing gradient step in eq. 10”.
> We have added the quoted statement to the revised manuscript for further clarification.
>
> **Why focus on stability**: The plasticity-stability dilemma is a well-established challenge in continual learning [A]. The standard neural network loss induces plasticity. Without measures to preserve stability, neural networks tend to be excessively plastic when introduced to new classes or tasks, due to the nature of the learning objective, and lack the stability necessary to prevent forgetting previous knowledge. Therefore stability preservation has been a key focus of continual learning approaches to prevent catastrophic forgetting [B, C]. The regularization provided by STAR attempts to balance this trade-off, and does not strongly prevent model plasticity and its ability to learn new tasks.
>
> [A] Mermillod et al. The stability-plasticity dilemma: Investigating the continuum from catastrophic forgetting to age-limited learning effects. Frontiers in psychology, 2013.
>
> [B] Mirzadeh et al. Understanding the role of training regimes in continual learning. Advances in Neural Information Processing Systems, 2020.
>
> [C] Lin et al. Beyond not-forgetting: Continual learning with backward knowledge transfer. Advances in Neural Information Processing Systems, 2022.

---

> > ### Comment · Reviewer_Rtfe · 2024-11-22
> >
> > Thank you for updating the paper with answers to my questions. The addition of 'worst' in Figure 3 adequately shows that the worst case solution is indeed reduced. Given the above response, I will raise my score, although one concern remains (see below), but it is less crucial.
> >
> > What still worries me a bit is that the paper does not show that the hypothetical situations in Figure 1 and 2 actually happen in practice. It is now clear that reducing the worst case scenario is beneficial, but the figures remain hypothetical. I would highly recommend to include these empirical measurements (even if it is in supplementary, I understand that there may not be enough space).

---

> ### Author Response · Authors · 2024-11-21
> **Gentle Reminder**
>
> We would like to thank you again for your constructive feedback, and give a gentle reminder that the discussion period will close in less than a week. We would be happy to further discuss any unresolved questions that you may have!

---

> ### Author Response · Authors · 2024-11-23
>
> We are currently in the progress of including a figure depicting empirical validation for fig 1. and fig 2. and we will certainly include it in the supplement of the camera ready version.
>
> We thank the reviewer for acknowledging our efforts and their positive comments. With that said, we would like to gently remind the reviewer to increase the score on openreview.

---

### Official Review · Reviewer_LJc1 · 2024-10-31

**Soundness:** 3
**Presentation:** 2
**Contribution:** 3
**Rating:** 6
**Confidence:** 4

**Summary:**

The paper focuses on improving rehearsal-based continual learning by stabilizing future predictions across the local neighborhood of the parameters. Specifically, it proposes a plug-and-play loss function, STAR, which applies parameter perturbation and reduces the KL-divergence between the model's predictions and those of its local parameters neighborhood. For each forward pass during training, a local neighbor of the current parameters is sampled, and this neighbor is perturbed by a single step of normalized gradient ascent to maximize the KL-divergence between the predictions of the model and the neighbor. Then, by combining the gradient of the KL-divergence between predictions with respect to the perturbed neighbor and the gradients of the rehearsal method's log-likelihood, the model parameters are updated cumulatively. This approach allows the models to learn a flat loss landscape, making the learned local parameter space less sensitive to future updates.

**Strengths:**

1. The paper is easy to follow and provides extensive experiments to show its plug-and-play effectiveness across different replay-based methods on small to large-scale datasets.

2. The authors have thorough experiments including ablation study, choice of buffer or current data for STAR loss and demonstration of distribution shift for seen tasks.

**Weaknesses:**

1. The paper lacks theoretical justification for why the method works, which could have further strenghtened the proposed method.

1. Minor Inconsistencies in Algorithm 1: (i) does not use epochs, (ii) two different hyper-parameters $\gamma$ and $\eta$ in equation (11) for perturbation coefficient (iii) $f$ used instead of $q$ in 345.

**Questions:**

1. Why is the perturbation ratio defined as the ratio of two norms in line 316 even though it is called and actually used as a hyper-parameter, as shown in table 7.

2. What is the essence for scaling gradients by the norm of weights in equation 11?

---

> ### Author Response · Authors · 2024-11-15
> **Initial Response**
>
> We thank the reviewer for praising our work as being “easy to follow”, and “thorough”.
>
> **Theoretical justification**: We would like to reiterate the connection of our formulation to forgetting based on empirical error (i.e. accuracy). Our method uses weight perturbation to reduce the KL-Divergence of the output with that of future output distributions. We refer to [A], which shows that empirical error can be bounded by some function of a classification calibrated loss function, such as KL-Divergence. The theoretical justification of weight perturbation, beyond our formulation, is a subject of extensive research: AWP [B] mentions a relationship with PAC-bayes generalization bounds but does not investigate thoroughly. Finally, the seminal work of [C] claims the PAC-bayes relationship is incomplete and provides convergence proofs for a worst-case perturbation method in the stochastic and non-convex scenario.
>
> **Perturbation norm ratio**: We would like to clarify and reiterate our formulation from the paper. The ratio gamma is not defined as the norm ratio, but **we design the perturbation $\delta$ such that the norm ratio  $\frac{\|\delta^{(l)}\|_2}{\|\theta^{(l)}\|_2}$ for each layer $l$ is approximately equal to $\gamma$**. The normalization is done on a per layer basis, and then scaled by the hyperparameter $\gamma$ for controlling the magnitude of the perturbation. This is motivated by different layers having different numerical distributions, as well as the scale invariance of the neural network layers [D] (i.e. multiplying the weights of one layer by a number and dividing the weights of the next layer by that number leads to the same network). We have added further clarification of this to the revised manuscript.
>
> Finally, we’d like to thank the reviewer for pointing out the algorithm inconsistency and we have fixed the notation. We would like to reiterate that the notion of time in our algorithm is not necessarily epoch and can be thought of as an optimization step.
>
> [A] Bartlett, Peter L., Michael I. Jordan, and Jon D. McAuliffe. "Convexity, classification, and risk bounds." Journal of the American Statistical Association 101.473 (2006): 138-156.
>
> [B] Wu, Dongxian, Shu-Tao Xia, and Yisen Wang. "Adversarial weight perturbation helps robust generalization." Advances in neural information processing systems 33 (2020): 2958-2969.
>
> [C] Andriushchenko, Maksym, and Nicolas Flammarion. "Towards understanding sharpness-aware minimization." International Conference on Machine Learning. PMLR, 2022.
>
> [D] Hao Li, Zheng Xu, Gavin Taylor, Christoph Studer, and Tom Goldstein. Visualizing the loss landscape of neural nets. Advances in neural information processing systems, 31, 2018

---

> > ### Comment · Reviewer_LJc1 · 2024-11-23
> >
> > Thank you to the authors for clarifying my questions and updating the manuscript. After reviewing your responses, I would like to respectfully maintain my initial rating, as I believe it aligns with my assessment of the manuscript's current state.

---

> ### Author Response · Authors · 2024-11-21
> **Genlte Reminder**
>
> We would like to thank you again for your constructive feedback, and give a gentle reminder that the discussion period will close in less than a week. We would be happy to further discuss any unresolved questions that you may have!

---

### Official Review · Reviewer_UhZb · 2024-11-03

**Soundness:** 3
**Presentation:** 3
**Contribution:** 3
**Rating:** 5
**Confidence:** 5

**Summary:**

In this paper, the authors mainly focused on maintaining the output distribution of previous models to prevent the catastrophic forgetting in rehearsal-based CL. To maintain the output distribution, the proposed method adopts not only the regularization between the future output and the output of the current model, but also minimizing the worst case version of this regularization. By doing so, the models can be updated toward the region in which the model outputs are well preserved. In the experiment, the authors show that the proposed method can strengthen the baselines, and also extensively conducted the ablation analysis.

**Strengths:**

Strengths

1. The viewpoint that the model should preserve the output distribution may be similar to the methods using the knowledge distillation in CL, the approach minimizing the worst case version of the regularization is novel. I think the critical difference between STAR and previous methods lies on the optimization scheme.

**Weaknesses:**

Weaknesses

1. I think the proposed method highly focuses on the stability of the model. If the number of incoming tasks is quite large (e.g. 50 tasks in split Omniglot), I wonder the proposed approach can still strengthen the baselines in the settings containing large number of tasks.

2. The authors said that this method does not assume any information on the task boundary. However, in the experiment, is it possible to consider the notion of epoch without the assumption on the task boundary? I know there is no terms on the task identifier in the formula, but I think the experiment setting is not consistent to the authors' argument. If the proposed method can cover any scenario in CL, does this methods also can work in single epoch setting?

3. The computational cost on optimizing the min-max loss is not negligible. I think it would be better to show the running time of this algorithm.

4. There is no experiments on large-scale dataset. Since the optimization procedure is much complex than previous methods, I wonder the proposed approach can be applied to much larger networks with large datasets

5. In terms of computing the gradient of Eq.8, the authors said that they assume the Hessian is identity matrix. However, I wonder using this gradient can find the minimum of the worst case loss function. In the all procedure, there are too many approximations to optimize the loss function.

**Questions:**

Already mentioned in the weaknesses section.

---

> ### Author Response · Authors · 2024-11-15
> **Initial Response**
>
> We thank the reviewer for their comments,suggestions, and constructive criticism. Here are our responses:
>
> **Scaling to more tasks**: We would like to kindly note that we have extensive experiments on three different datasets, which are standard and adopted for the works of [A,B]. As for Omniglot (50 tasks), while increasing the number of tasks naturally makes the CL scenario more challenging, we have no reason to believe our method would be particularly disadvantaged compared to other CL/enhancement methods. Furthermore, Omniglot is a one-shot learning dataset with limited data samples (~1600) and additional challenges which we believe to be outside the scope of this paper.
>
> **Notion of epochs**: We would like to reiterate that our formulation does not necessarily equate time to an epoch, but the notion of time can be considered to be a single optimization step, making it easily extendible to single-epoch/online scenarios. Our optimization technique is performed on each batch regardless of how many epochs there are. **As a proof of concept, we’ve added a limited experiment at the end of this comment (table 1).** We perform single-epoch training of the S-CIFAR100 dataset with 3 training steps per batch for all settings and average over 5 seeds. The results suggest that STAR leads to performance improvements in the single-epoch setting as well. With that said, the online CL setting comes with its own challenges and may require additional considerations (such as additional augmentations and repeated rehearsal [E]), there are a variety of works tackling the online CL scenario, such as [C] which we mention in our related works. In our added results, we experiment with repeated rehearsal as well as without it. We believe a dedicated, thorough extension of our method to the online scenario could be a task for future works.
>
> **Running time**: We thank the reviewer for the suggestion and will report the running times for table 1 in the revised version in the supplement. We would like to note that while our method does indeed involve extra computational expense, it is not a prohibitive cost (only two additional forward/backward steps per batch) and leads to an improvement in performance. Improving the efficiency of our method (and perhaps that of other worst-case weight-perturbation methods) can be the subject of future works.
>
> **Large-scale datasets and models**: We utilized traditional CL benchmark datasets and would like to note that we conducted experiments on miniImagenet, which is a benchmark large-scale CL dataset. The additional computational cost introduced by STAR is not prohibitive for larger models as it only adds two additional forward/backward passes during training.
>
> **Hessian approximation**: The approximation utilized is a first-order approximation of the KL loss (eq. 7).  This approximation is widely used by existing works in the literature which leverage worst-case weight perturbation [C,D], and works well in practice, as shown in our experiments. Furthermore, the work of [F] provides convergence guarantees for Sharpness Aware Minimization which utilizes a first-order approximation on worst-case weight perturbation.
>
> Table1. Proof of concept results for single-epoch training on S-CIFAR100, all methods train for 3 steps on each batch. Averaged over 5 seeds.
>
> ---
> | Buffer Size | 2000      | 5000      |
> |-------------|-----------|-----------|
> | ER          | 33.74     | 41.38     |
> | ER + STAR   | **34.69** | **42.36** |
> ---
>
> [A] Bonicelli, Lorenzo, et al. "On the effectiveness of lipschitz-driven rehearsal in continual learning." Advances in Neural Information Processing Systems 35 (2022): 31886-31901.
>
> [B] Wang, Zifeng, et al. "DualHSIC: HSIC-bottleneck and alignment for continual learning." International Conference on Machine Learning. PMLR, 2023.
>
> [C] Foret, Pierre, et al. "Sharpness-aware minimization for efficiently improving generalization." arXiv preprint arXiv:2010.01412 (2020).
>
> [D] Wu, Dongxian, Shu-Tao Xia, and Yisen Wang. "Adversarial weight perturbation helps robust generalization." Advances in neural information processing systems 33 (2020): 2958-2969.
>
> [E] Zhang, Yaqian, et al. "A simple but strong baseline for online continual learning: Repeated augmented rehearsal." Advances in Neural Information Processing Systems 35 (2022): 14771-14783.
>
> [F] Andriushchenko, Maksym, and Nicolas Flammarion. "Towards understanding sharpness-aware minimization." International Conference on Machine Learning. PMLR, 2022.

---

> ### Author Response · Authors · 2024-11-21
> **Gentle Reminder**
>
> We would like to thank you again for your constructive feedback, and give a gentle reminder that the discussion period will close in less than a week. We would be happy to further discuss any unresolved questions that you may have!

---

> > ### Comment · Reviewer_UhZb · 2024-11-25
> > **Reply to the comment**
> >
> > Thank you for your effort on answering my questions. However, the problems are still not resolved yet.
> >
> > In case of the large scale dataset experiment, I don't think the mini-Imagenet dataset represents the large-scale dataset. As the author said that STAR contains additional forward / backward procedure, the additional computational cost on large scale experiment would not be negligible. In many class-incremental learning scenarios, the datasets like ImageNet-1K are widely used, and the overall tendency on large scale experiment is totally different from small-scale experiments.
> >
> > For the large number of tasks, I still wonder STAR can generalize well when the number of tasks are large. Since STAR mainly focuses on the stability of CL algorithm, STAR may fail to have high plasticity on proceeding tasks.
> >
> > Since most of the problems are still unresolved, I will keep my score.

---

> > > ### Author Response · Authors · 2024-11-25
> > > **Thank you for your review**
> > >
> > > We thank you for your review, appreciate that you engaged in discussion and respect your decision to maintain your score.
> > >
> > > We would like to emphasize that existing methods in the literature, e.g. our competing methods such as LiDER or DualHSIC, and the baselines over which we applied STAR such as X-DER, use datasets of the same scale or smaller including different subsets of Imagenet like tiny Imagenet or mini Imagenet. To the best of our knowledge, evaluation on ImageNet-1K  is not standard and requires extensive computational resources.

---

### Official Review · Reviewer_yANV · 2024-11-04

**Soundness:** 3
**Presentation:** 2
**Contribution:** 2
**Rating:** 8
**Confidence:** 4

**Summary:**

The paper introduces STAR, a plug-and-play loss component designed to enhance rehearsal baselines by addressing potential forgetting from future parameter updates. Since future parameters are unknown, STAR estimates forgetting through a surrogate measure: i.e., capturing the worst-case perturbation of current parameters within their local neighborhood. Substantially, the authors argue that making the model resilient to perturbations (with a dedicated loss components) helps in reducing the future forgetting. Ideally, STAR is evaluated across three datasets, comparing rehearsal baselines i) with and without its application, and ii) against other state-of-the-art plug-and-play components.

**Strengths:**

1) Tackling the problem of future forgetting by acting on the current task is novel and interesting;
2) the ablations and exploratory experiments are concise yet to the point;
3) leveraging straight weight perturbation as a regularizer when training in continual learning is compelling, although similar in spirit to [1].

[1] Lorenzo Bonicelli, Matteo Boschini, Angelo Porrello, Concetto Spampinato, and Simone Calderara. On the effectiveness of lipschitz-driven rehearsal in continual learning. Advances in Neural Information Processing Systems, 35:31886–31901, 2022.

**Weaknesses:**

1) While an improvement over existing rehearsal baselines is interesting, its appeal is limited as these baselines have largely been surpassed by prompting approaches. Indeed, some of these techniques [1, 2] now represent the state of the art in Continual Learning;
2) while the improvements on Split-CIFAR10 are solid, those on Split-CIFAR100 and Split-miniImageNet (Table 2) are far less noticeable and sometimes absent;
3) the results in Table 1 for Split-CIFAR100 seems to be different for X-DER [3] to what reported in the original paper. This hinders a good evaluation, as the original results (reported in [3]) surpass those of X-DER equipped with the proposed methodology.

Generally, I feel this work is incremental w.r.t. LiDER [4] in its idea. Also, the improvement w.r.t. other plug-and-play techniques appears not significant enough.

Some minor issues that did **not** affect my evaluation:
 - In the related works section, regularization-based methods are listed twice.
 - In the line preceding Eq. 4, “f” should be “f(x).”
 - For the gradient ascent step, eta seems to be used in place of gamma (as in Figure 2).
 - In the explanation of the gradient ascent, the equivalence of eta within the i.e. parentheses appears incorrect.
 - In Algorithm 1, “f” is used instead of “q” in the STAR gradient.


[1] Smith, James Seale, et al. "Coda-prompt: Continual decomposed attention-based prompting for rehearsal-free continual learning." Proceedings of the IEEE/CVF Conference on Computer Vision and Pattern Recognition. 2023.

[2] Wang, Liyuan, et al. "Hierarchical decomposition of prompt-based continual learning: Rethinking obscured sub-optimality." Advances in Neural Information Processing Systems (2024).

[3] Boschini, Matteo, et al. "Class-incremental continual learning into the extended der-verse." IEEE transactions on pattern analysis and machine intelligence (2022).

[4] Bonicelli, Lorenzo, et al. "On the effectiveness of lipschitz-driven rehearsal in continual learning." Advances in Neural Information Processing Systems (2022).

**Questions:**

None

---

> ### Author Response · Authors · 2024-11-15
> **Initial Response**
>
> We would like to thank the reviewer for their comments and for describing our method as “novel” and “interesting” and our evaluations as “concise and to the point”.
>
> **LiDER**: We would like to start our response by highlighting the major difference between our work and that of LiDER [A]. First, LiDER focuses on regularizing the Lipschitz constant of the neural network, while STAR (ours) optimizes the KL-divergence of the output distribution between current parameters and the potential future parameters. Second, and more importantly, **computing the Lipschitz constant is non-trivial** and requires approximations and relaxations based on the activation functions and possibly the architecture of the network. To quote the authors of LiDER in the limitations section **“our approximation cannot be applied to not-Lipschitz continuous layers (e.g., cross-attention)”**. STAR, on the other hand, is general to probabilistic modeling of the output classes and can be used with attention-based architectures, e.g. the transformer.
>
> **Prompting-based methods**:  First, the usage of prompting methods in CL requires pre-trained models, often with a large number of parameters, and furthermore, comes with certain limitations in terms of expressiveness [C]. Our method, on the other hand, is applicable to a general training scheme. Second, we would like to note that our method is a complementary approach that can also be combined with prompting methods with the inclusion of a small rehearsal buffer (as is done in some existing CL prompting works [B]).
>
> **Results on CIFAR100 and miniImageNet**: We would like to note the results in table 1 and that our improvements over most existing baselines are significant (up to a ~10% increase in accuracy across the two datasets). Regarding the results in Table 2, out of the 12 experiment settings presented for CIFAR100 and miniImagenet in Table 2, we achieve best or second best accuracy compared to competing enhancement methods for 10 of them.
>
> **Mismatched results in Table 1**: We thank the reviewer for the keen eye and noticing the difference, however, we use the same hyperparameters as [A, D] and reported the same results for the X-DER [E] (meaning [A,D] also suffer from this inconsistency). Upon further investigation, we suspect this is due to using a different batch size (64) than that of the original paper (32). We have rerun the experiments with the original batch size and have updated the results in the revised manuscript in Table 1. (Note that the batch size used for this experiment is not reported in [E]).
>
> **Minor errors**: Thank you for indicating these, we have revised them in the manuscript.
>
>
> [A] Bonicelli, Lorenzo, et al. "On the effectiveness of lipschitz-driven rehearsal in continual learning." Advances in Neural Information Processing Systems 35 (2022): 31886-31901.
>
> [B] Wang, Zifeng, et al. "Learning to prompt for continual learning." Proceedings of the IEEE/CVF conference on computer vision and pattern recognition. 2022.
>
> [C] Wang, Yihan, et al. "Universality and limitations of prompt tuning." Advances in Neural Information Processing Systems 36 (2024).
>
> [D] Wang, Zifeng, et al. "DualHSIC: HSIC-bottleneck and alignment for continual learning." International Conference on Machine Learning. PMLR, 2023.
>
> [E] Boschini, Matteo, et al. "Class-incremental continual learning into the extended der-verse." IEEE transactions on pattern analysis and machine intelligence 45.5 (2022): 5497-5512.

---

> ### Comment · Reviewer_yANV · 2024-11-19
>
> Thank you to the authors for taking the time to thoroughly address my questions. I appreciate that they corrected the results in Table 1 with the right batch size, and suggest to check all the other hyperparameters for a fair comparison.
>
> I still emphasize that the results in Table 2 are not particularly impressive. However, Table 1 demonstrates a notable improvement over other state-of-the-art replay baselines, and I acknowledge that prompting techniques, which depend on pre-trained models, are beyond the scope of this work.
>
> After the last changes, I now believe this is a strong submission and have accordingly raised my score to 8.

---

> > ### Author Response · Authors · 2024-11-21
> > **Thank you for the review**
> >
> > We have double-checked the remaining experiment settings to ensure the correct hyper-parameters are used and that there are no significant numerical discrepancies with existing works.
> >
> > Finally, we would like to thank the reviewer again for confirming our efforts towards a fair comparison by updating the results, recognizing the significance of the improvements in Table 1., and for considering our paper to be novel ,interesting, and a strong submission.

---

### Author Response · Authors · 2024-12-02
**Discussion Summary**

We would like to thank all the reviewers for the constructive feedback, and especially for engaging in discussion. We sincerely believe your feedback has made this work better.
As the discussion window comes to a close, we would like to summarize below the changes made during the discussion period. We’ll continue to incorporate any remaining suggestions in the camera-ready version upon acceptance.

Summary of changes:
- Added a ‘worst’ case scenario to Fig. 3 to ensure the worst case KL-Div is being reduced.
- Updated results of X-DER in Table 1. for CIFAR100 to be consistent in hyperparameters with the original work.
- Correction of notational inconsistencies in Alg. 1.
- Running times for the algorithms in Table 1 were added to the supplementary.
- Removed repeated mention of ‘regularization-based’ methods from the related works section.

---

### Meta-Review · Area_Chair_5EFs · 2024-12-20

**Metareview:**

This paper tackles rehearsal-based methods for continual learning by changing the loss function to have (an approximation to) a KL divergence. The idea is relatively simple (which I view as a strength), although similar in spirit to related work both mentioned in the paper and by reviewers. There are experiments across benchmarks and datasets to show empirical improvements.

Reviewers agree that the method is well-motivated (aside from a concern from Reviewer Rtfe about Figures 1 and 2, which the authors promised to address in the future version), and performs well on the benchmarks tested on. All reviewers except for UhZb thought that the empirical results were sufficient and good.

That said, during reviewer discussion, reviewers agree that the contribution is relatively incremental (ie not a groundbreaking idea or method), an assessment I agree with. Reviewer UhZb has concerns about the maximum size and task length in the experiments. This is very fair, and I think the paper would be stronger even with an experiment with much longer task sequence. However, in my opinion, this is not necessary for acceptance to ICLR.

**Additional Comments On Reviewer Discussion:**

Reviewer yANV was convinced by author rebuttal and increased their score to accept, and argued for acceptance in reviewer discussion. Reviewer LJc1 also felt that their concerns were addressed during the rebuttal, but that the contribution was too incremental to warrant a rating above 6. Reviewer Rtfe raised some points I agree with, such as that perhaps "the results are explained by a different hypothesis, e.g. reduced overfitting on the memory". They said they will increase their score but did not, and did not partake in reviewer discussion either. The authors promised to include empirical versions of Figs 1 and 2 in a camera-ready version, which I think is a relatively minor point, but would help understanding and intuition of the method.

Reviewer UhZb's main concern is about the lack of larger scale datasets and longer task sequences. On balance, I find myself agreeing with the other reviewers and the authors, that the curent empirical results are enough for a conference paper, like in other papers. The paper would undoubtedly be stronger if it had a larger-scale dataset or a longer task sequence.

---

### Decision · Program_Chairs · 2025-01-22

Accept (Poster)